# Use of Digital Biomarkers from Sensing Technologies to Explore the Relationship Between Daytime Activity Levels and Sleep Quality in Nursing Home Residents with Dementia: A Proof-of-Concept Study

**DOI:** 10.3390/s25216635

**Published:** 2025-10-29

**Authors:** Lydia D. Boyle, Monica Patrascu, Bettina S. Husebo, Ole Martin Steihaug, Kristoffer Haugarvoll, Brice Marty

**Affiliations:** 1Centre for Elderly and Nursing Home Medicine, Department of Global Public Health and Primary Care, University of Bergen, Årstadveien 17, 5009 Bergen, Norwaybettina.husebo@uib.no (B.S.H.); brice.marty@uib.no (B.M.); 2Neuro-SysMed, Department of Neurology, Haukeland University Hospital, Jonas Liesvei 65, 5021 Bergen, Norway; kristoffer.haugarvoll@helse-bergen.no; 3Complex Systems Laboratory, University Politehnica of Bucharest, Splaiul Independentei 313, 060042 Bucharest, Romania; 4Department of Internal Medicine, Haraldsplass Deaconess Hospital, Ulriksdal 8, 5009 Bergen, Norway; ole.martin.steihaug@haraldsplass.no; 5Neuro-SysMed, Department of Clinical Medicine, University of Bergen, Jonas Liesvei 65, 5021 Bergen, Norway

**Keywords:** dementia, technology, sensors, nursing home, sleep quality, physical activity

## Abstract

Inactivity and increases in psychological and behavioral symptoms are common for people with dementia, and current assessment relies on proxy-rated tools. We investigate the feasibility and adherence of the use of sensor technology by exploring the relationship between daytime activity and sleep quality. For a total of 42 day–night data pairs in nursing home residents with dementia (N = 11), Garmin Vivoactive5 and Somnofy monitored continuous physical activity levels, sleep efficiency (SE), sleep score, sleep regularity index (SRI), and wake after sleep onset (WASO). Using the Spearman coefficient, we explored correlations between digital and proxy-rated tools (Personal Self Maintenance Scale (PSMS) and Neuropsychiatric Inventory–Nursing Home version (NPI-NH)) and the relationships between the digital biomarkers (SE, SRI, WASO, sleep score, physical activity). Participants (mean age 84 years) had moderate to severe degrees of dementia. Daytime activity levels correlated to sleep quality parameters WASO (−0.34, *p* = 0.03), and SRI (0.43, *p* = 0.01), and traditional sleep measures were associated with digital biomarkers (WASO/NPI-NH-K, *p* = 0.03). We found a relationship between daytime activity and sleep quality; however, the bidirectional relationship remains ambiguous and should be further investigated. The use of sensing technologies for people with dementia residing in a nursing home is feasible, although not without limitations, and has the potential to identify subtle changes, improving clinical assessment and the corresponding care recommendations.

## 1. Introduction

Decreased physical activity and increased sleep disturbances are common in nursing home residents with dementia and are often linked to polypharmacy and higher mortality [1]. Current methods for monitoring these changes are often unreliable, as they rely heavily on proxy-rater assessment. Given the high prevalence of dementia in Norwegian nursing homes (at least two-thirds of residents) [2,3] and the global rise in dementia cases (an estimated 139 million worldwide by 2025), there is a critical need for improved, technology-driven monitoring to increase more precise diagnostics and treatments [1].

People with dementia exhibit significant daytime inactivity, which negatively impacts their daily living and quality of life [4,5]. While some activity peaks in the afternoon and early evening [5], they largely experience sedentary behaviors such as sitting, watching TV, or sleeping, eating, and drinking as the primary daily physical focus [2]. To create effective interventions, accurate assessment of physical activity levels is crucial, and the combined use of digital biomarkers (using multiple sensing technologies) with traditional methods has the potential to improve understanding of the relationship between daytime activity and sleep quality.

Sleep quality significantly declines with age and is particularly problematic for people with dementia, often leading to the exacerbation of behavioral and psychological symptoms in dementia (BPSD), such as agitation, psychosis, and depression [6,7,8]. At least four age-related changes in sleep behavior are described in the previous literature; decreased sleep efficiency (SE), which is the proportion of time spent asleep while in bed; reduced sleep duration, signaling the reduction in total sleep time (TST); less deep sleep (slow-wave sleep), impacting the restorative quality of sleep; and increased wake after sleep onset (WASO) with more frequent awake time during the night [7]. While polysomnography is the gold standard for measuring sleep quality (sleep efficiency, duration, latency, stages, etc.) [9], its application in nursing homes is limited, necessitating the use of proxy-rated questionnaires such as the Neuropsychiatric Inventory—Nursing Home version (NPI-NH; item for sleep quality, section K). However, the assessment of sleep by a nighttime nurse may not reflect the real clinical situation, as demonstrated by earlier research [10].

Sensing technologies used for human activity and behavioral recognition (HAR), such as smartwatches and contactless radar, are widely available with the potential to further our understanding of activity and sleep disturbances. Our umbrella review identified nine recent systematic reviews describing the use of sensing technologies for HAR in people with Parkinson’s disease and dementia. We found that sensing technologies for use as assessment tools are promising; however, they are diversely applied and currently not well-integrated for clinical use, especially concerning the behavioral symptoms of dementia [11].

An Australian study by Moyle et al. (2017) included people with dementia for a 24 h observation of activity and sleep, using traditional questionnaires and a triaxial accelerometer (armband), and demonstrated high inactivity during daytime hours and an average of seven hours total sleep time [5]. Adherence to the wearable showed large variations, with about 50% data acquisition for final analysis [5]. A case series study from the United States by Au-Yeung et al. (2022) [12] used actigraphy to follow a person with dementia during the last months of their life. Their findings highlight the potential to apply sensing technologies for assessing activity levels in connection with psychotropic drug use and subsequent changes at the end of life [12]. Moreover, the methodology for traditional vs. technical assessment was investigated in nursing home residents with dementia, underlining the limited validity of traditional tools compared to technology-based solutions [10].

This explorative proof-of-concept study aims to examine the feasibility for the clinical use of digital biomarkers from sensing technologies in nursing home residents with dementia, through an investigation of the relationship between daytime activity levels and sleep quality. The Digital Phenotyping for Changes in Activity at the End of Life in People with Dementia study (DIPH.DEM) is an observational pilot study using sensing technology and traditional assessment tools in dementia and digital biomarkers. The following research questions will be explored within this connected proof-of-concept study:What is the adherence, feasibility, and quality of digital data acquired by Garmin Vivoactive5 and Somnofy?To what extent are the selected digital biomarkers for activity and sleep related to the selected clinical questionnaires in dementia?Which digital sleep parameters show the strongest correlations with daytime activity levels in nursing home residents with dementia?

## 2. Materials and Methods

### 2.1. Design and Study Population

This proof-of-concept study is a part of the exploratory and observational pilot study, DIPH.DEM, exploring the capabilities of using digital biomarkers in the observation of relationships between daytime activity levels and sleep quality during the end of life in people with dementia. DIPH.DEM (REK 634938), an external pilot study, was designed as a smaller scale study to Decoding Death and Dying in People with Dementia by Digital Thanotyping (5-D) (REK 657596, NEM 2023/166), and intended to be an informative study for improvement of the quality and efficacy of 5-D. Because the main purpose of a pilot study is not hypothesis testing, a sample size calculation was not conducted; however, a sample size of 10–12 participants [13,14] and a minimum of 50 data-point observations for digital biomarker analysis [15] were deemed acceptable for addressing the study’s specific research questions and in alignment with prior methodological recommendations for pilot studies and investigations involving digital biomarkers. We further based the sample size upon an estimate of 5–10% of the anticipated total sample size of the larger study. Because the use of digital phenotyping is not yet common in the intended population, DIPH.DEM’s purpose was two-fold: to improve the 5-D study recruitment methods and design, and to assess the overall feasibility of the use of sensing technologies and digital biomarkers for assessment of activities and behavioral symptoms, such as sleep, in people with dementia residing in a nursing home.

A proof-of-concept methodology was used in this study for the investigation of the feasibility to use sensing technologies to measure physical activity and sleep quality for people with dementia residing in a nursing home, and as a prerequisite to the larger explorative observational study (5-D) involving similar methods. Participants (N = 11) were recruited through two nursing home wards in Bergen, Norway. Residents were eligible for inclusion if they were 65 years or older, had lived in the nursing home for >6 weeks, and had a diagnosis of dementia or moderate-to-severe cognitive impairment assessed by the Clinical Dementia Rating assessment (CDR). Exclusion criteria were presence of delirium at baseline assessment and a current life expectancy of less than 6 weeks (suggested by the multidisciplinary nursing home team).

### 2.2. Clinical Assessment Tools

Prior to data and demographics (age, gender) collection, training was provided by the researcher to the healthcare professionals at the nursing home: usually the primary nurse working with the participant daily. The training included verbal and written information about the use of the traditional proxy-rater assessment tools and sensing technologies.

The Clinical Dementia Rating questionnaire (CDR, range 0–3) is a widely accepted global scale, developed to clinically identify the presence of dementia and stage its severity (score of >0.5—cognitive impairment/likely dementia; 1–mild, 2–moderate, 3–severe dementia) [16]. Within our study, participants were included if they had a score equal to or greater than one on the CDR. The 4AT Test (range 0–12) is a screening tool for delirium and consists of four questions evaluating alertness, cognitive function, and acute change, with a score of four or above indicating potential delirium [17]. The General Medical Health Rating (GMHR) (range 1–4) assesses the number and impact of comorbidities and concomitant medication use (1 = very good; 2 = good; 3 = moderate, and 4 = bad health) [18].

The Neuropsychiatric Inventory—Nursing Home version (NPI-NH) consists of twelve symptoms [19] to assess behavioral disturbances in dementia, such as delusions, hallucinations, agitation, aggression, depression, anxiety, euphoria, apathy, irritability, aberrant motor behavior, nighttime behavior, and eating behavior (range 0–12 in each item; 0–144 total score). The Norwegian version of NPI-NH was found to be reliable and valid [20,21]. For our study, we focus on results related to sleep quality and behaviors (NPI-NH-K). The Physical Self Maintenance Scale (PSMS) assesses activity levels and the adverse effects of inactivity on the functional activities of daily living [22,23]. The questionnaire includes six items with five possible answers, including functional activities of ambulation, bathing, grooming, dressing, toileting, and feeding (range 6 to 30), with higher scores indicating a greater need for support (6 = independent, 7–12 = supervision- minimum assist, 13–18 = minimum assist, 19–24 = moderate assistance, 25–30 = max assistance). All assessment tools have been previously validated and were available in the Norwegian language.

### 2.3. Sensing Technologies

Garmin Vivoactive5 (4 GB) firmware version 11.14 (Garmin Ltd., Switzerland) is a wrist-worn commercial smartwatch with acceleration and heart-rate sensors that is able to collect information about day- and nighttime physical activity levels [24]. The watch has been validated in older adults < 65 years [24]. The device was chosen based on established evidence-based guidelines and frameworks [25]. Based on initial feedback from the participants and healthcare team, the smartwatches were programmed with a realistic clock face; no sound, alarms or vibration; low lighting; and locked features, so that tampering with the settings would not occur. For research purposes, smartwatches were placed on the participants by the researcher on day one of data collection, removed and charged on day four by the onsite healthcare contact at the nursing home, and placed back on the participant afterwards (1–2 h missing data), prior to being collected again by the researcher on day seven. To improve generalizability, raw data parameters were synced, paired, and retrieved using Fitrockr [26], a cloud-based data storage service (using latest firmware updates from Garmin Ltd.) based in Berlin, Germany, and acceleration was collected at 25 Hz. Participants were asked to wear the smartwatch on their preferred wrist continuously over the data collection period of 7 days, including when bathing and sleeping.

Somnofy (Vital Things AS, Trondheim, Norway) version 0.7 with sleep algorithm 1.0 is a contactless radar technology that is able to collect information about sleep cycles and different elements of quality of sleep; sleep efficiency (SE), sleep score, wake after sleep onset (WASO), sleep regulatory index (SRI), total sleep time (TST), sleep latency, total time in bed, and no presence (time out of bed) [27]. The average sampling rate for Somnofy is 23.8 Hz, with 30 s epochs mimicking gold standard Polysomnography (PSG). Somnofy has been validated against PSG in an adult Norwegian population (19–61 years, mean age 28.9 years, SD 9.7) [27], with epoch-by-epoch analyses reflecting a sensitivity and specificity of 0.97 and 0.72 for Somnofy, compared to 0.99 and 0.85 for PSG. For study purposes, Somnofy was installed by the researcher and placed on the wall above the bed in each room several days prior to the data collection start, to anticipate technical challenges. During installation, all lights and audio alerts were turned off, and the device was connected to a power source. Once the participant was within sensor range (1–3 m), a sleep session began. Data gathered using Somnofy was collected continuously for six nights.

## 3. Data Quality, Extraction, and Preprocessing

### 3.1. Adherence, Feasibility, and Residents’ Safety

The quality of the data was assessed by observation of the amount of missing data (completeness), adherence to use of the smartwatch by the user (total wear time), accuracy, accessibility (raw data), clarity, and comparability. We used common dimensions of established quality data frameworks to guide the assessment of data quality within this study [28]. Total adherence was assessed using wear time of the Garmin Vivoactive5 over the continuous data collection period, excluding the initial extraction of the first and last partial days when the devices were donned and doffed by the researcher, for five days total per participant. Feasibility was determined based on the researcher’s observation and evaluation of the technology, via informal questions to the primary nurse during and immediately post data collection. Training and written instructions for the healthcare representatives included components of feasibility in the instruction for the daily inspection of the participant’s skin (comfort and injury prevention), adherence to continuous donning of the smartwatch, and communication during the seven-day period about any behaviors which could affect the data quality, such as the participant not sleeping in their own bed, BPSD related challenges, or periodic removal of the smartwatch. Informal post data collection questions were asked of the primary nurse by the researcher concerning any feedback or challenges they experienced during data collection.

### 3.2. Acceleration

Acceleration data were collected for a total of 7 days and for 24 h each day. Raw acceleration data of each participant were retrieved using the Fitrockr hub and downloaded from the platform in the comma-separated values (CSV) format. The extracted data were imported and converted in MATLAB-compatible types and based on the Unix time timestamps. Unix time segmented the data into 14 h periods (07:00:00–20:59:59) representing daytime; the first and last partial days were excluded to only consider full-day recordings. A full-day recording was considered to be a data profile containing a minimum of 21 h of data, accounting for normal technical pauses in data collection, such as charging and periodic removal of the device.

Euclidean norm minus one (ENMO) is the vector magnitude of the three raw signals (*x*, *y*, *z*) minus 1. Periods of 1 min were used during daytime and daily 24 h (07:00:00–06:59:59) measurements (mean, median, SD). ENMO is calculated by summing the squared acceleration of each of the three accelerometer axes at each time point and then subtracting the gravitational component (1 g = 9.81 m/s^2^).$$ENMO_{i}=\sqrt{{{x\left(t\right)}_{i}}^{2}+{{y\left(t\right)}_{i}}^{2}+{{z\left(t\right)}_{i}}^{2}}-1000$$
where ${x\left(t\right)}_{i}$, ${y\left(t\right)}_{i}$ and ${z\left(t\right)}_{i}$ are accelerations on the three axes of the *i*th datapoint of the sample considered in milli-g.

The mean amplitude deviation (MAD) [29] is a comparable universal metric that uses ENMO in its calculation, utilizing the mean signal value to reduce the effect of the constant gravitational acceleration. MAD is calculated as follows:$$MAD(t_{0};H)=\frac{1}{H}\Sigma_{h=0}^{H-1}|r(t_{0}+h)-\bar{r}(t_{0};H)|$$
where *H* is the size of the considered epoch, $r\left(t\right)\;$ is the Euclidean norm of the acceleration,$$r(t)=\sqrt{{{x\left(t\right)}_{i}}^{2}+{{y\left(t\right)}_{i}}^{2}+{{z\left(t\right)}_{i}}^{2}}$$

And $\bar{r}(t_{0};H)$ the average Euclidean norm in the epoch:$$\bar{r}(t_{0};H)=\frac{1}{H}\Sigma_{h=0}^{H-1}r(t_{0}+h)$$

The unit of the MAD, like ENMO, is milli-g, and 5 s epochs were used for both metrics.

### 3.3. Sleep Digital Biomarkers

The sleep quality digital biomarkers were collected for a total of six nights. Sleep biomarker data were retrieved using the Somnofy platform, following the company provided guidelines, and downloaded in Excel Binary File Format (.XLSX). Data were manually inspected to exclude daytime sleeping (naps) and considered nighttime to be the period of 21:00:00–06:59:00. This timeframe was chosen to reflect the nightly distribution of sleep medications and bedtime routines at the nursing home at approximately 21:00 each evening. Data in the designated nighttime data profile was removed if the participant had an empty or partial recording that was less than 2 h in duration. Extracted data were imported and converted to MATLAB-compatible types.

SE is calculated by dividing the time asleep (seconds) by the total time in bed (seconds) (×100) and was provided in Somnofy’s base data. The total time asleep includes the following: sleep onset latency (SOL) + TST + WASO + time attempting to sleep after the final awakening [30]. SE ranges from 0 to 100%, where a score of 100% is a perfect score, indicating no awake time between sleep onset and termination. A previous study [31] including community-dwelling, inactive, frail older adults (>65 years) found a mean SE of 84%. WASO is the total number of minutes that a person is awake after having initially fallen asleep (sleep latency). It has an inverse relationship with SE; as WASO increases, SE decreases. Normal ranges for WASO measured with gold standard polysomnography in healthy adults aged 61–73 years ranges from 77.3 ± 35.6 min per night [32]. SRI uses a metric that compares the similarity of sleep patterns day-to-day (i.e., 7 days/6 nights) and calculates the averages in sleep–wake states over all epoch pairs, separated by 24 h (100 = perfect pattern; 0 = random patterns) [33].

SRI scores were calculated using MATLAB R2023a, with the following formula:$$SRI=-100+200\left(1-\frac{1}{N_{v}}\sum_{i=1}^{N}\left|s_{i}-s_{i+C}\right|\right)$$
where *N* is the number of available days, $N_{v}$ the number of available epoch-by-epoch comparisons, $s_{i\;}$ is the sleep–wake state ($s_{i}$ = 1 for wake, $s_{i}$ = 0 for sleep), C is the number of epochs within one 24 h interval, $s_{i}$ ≠ Not Available (NA), and $s_{i+C}$ ≠ NA. Excluded epochs were represented by $s_{i}$ = NA.

### 3.4. Statistical Analysis

To test the normality of our data distribution, the one-sample Kolmogorov–Smirnov Test was used, under the null hypothesis that the samples come from a standard normal distribution. For all parameters, both activity (EMNO/MAD daytime) and sleep (SE, sleep score, SRI, and WASO), the null hypothesis was rejected, showing that none of the samples followed a normal distribution. Therefore, because the data were non-parametric, skewed, and non-linearly distributed, Spearman rank correlation tests were used to display potential correlations between all variables, using bias-corrected and accelerated bootstrapping techniques (BCa) due to the sample size (5000 replications): reporting correlation coefficient (rho), 95% confidence intervals (CI), standard error, and probability (*p*-value). The hypothesis of no correlation was tested against the alternative hypothesis of a nonzero correlation, and results were considered to be statistically significant at *p* < 0.05. Cohen’s guidelines [34,35] were used as a general reference for interpreting effect sizes: 0.10–0.29 (small), 0.30–0.49 (medium), and ≥0.50 (large). Scatter plots were used to demonstrate the relationship between daytime activity and sleep quality variables. Descriptive statistics and narrative synthesis are provided for the group, including mean, median, min, max, and standard deviation for all relative variables. We note that the reader should interpret these results with caution, based upon the relatively low sample size, which can affect the interpretation of the results. However, we provide an analysis of the digital biomarkers, featuring, as recommended [15], more than 50 digital observations and expanded statistical and descriptive measures, in efforts to improve generalizability for future studies. Analyses were conducted using MATLAB R2023a and STATA SE18.5.

### 3.5. Ethics

Verbal and written informed consent were obtained in direct conversation with the residents who demonstrated sufficient capacity to consent. For those lacking this ability, we obtained consent from the resident’s legal guardian, usually a family member, after explaining the aims and protocol of the study. The DIPH.DEM study was approved by the Regional Committee for Medical and Health Research Ethics (REK) in Norway in October 2023: approval number 634938.

## 4. Results

From February 2024 to May 2024, eleven participants were recruited; one resident died directly after baseline assessment. The mean age was 84.0 years (SD ± 6.1) and eight participants were female (Table 1). The participants had been living in the nursing home for an average of 1.5 years. All participants had dementia with a median CDR score of 2.2 (1–3), indicating a median level of moderate dementia within the group; according to CDR cut-offs, seven of the participants were classified as having moderate-to-severe dementia, and four as having mild dementia. Chronic disease other than dementia within the group included Alzheimer’s disease, Parkinson’s disease, diabetes, and heart disease. According to the GMHR, comorbidities within the group were “moderate”, indicating that most had >3 unstable conditions, or several stable conditions that were chronic, requiring several daily medications to manage symptoms. The mean number of psychotropic drugs per participant being used was 1.9 (range of 0–5) and the total polypharmacy mean was 7.6 (range of 2–13). Participants with a score of ≥2 on the CDR (moderate to severe dementia, N = 7) had a similar (mean) use of total daily medications of 7.7, compared to those scoring 1 on the CDR (mild dementia, N = 4) with 7.5. Meanwhile, nightly prescribed psychotropic medications were two times greater (2.5:1.25) in the participants with a CDR score of ≥2 (moderate-to-severe dementia). As CDR scores increased, so did the number of total and psychotropic medications. Assessed by NPI-NH, five had symptoms of irritability and 36/45/27% had symptoms of depression, agitation, and anxiety, respectively. Most participants were classified as needing minimum-to-moderate assistance with daily activities, scoring between 16 and 23 on the PSMS, which indicated a moderate level of functional impairment within the group (Table 1). The highest areas of impairment were dressing, self-care, and bathing. Sleep disturbances were observed in six of the participants (NPI-NH-K). The group’s mean SE at 70.8% was approximately 13 percentage points (16%) less than the average reported in similar populations (Table 1 and Table 2).

### 4.1. Adherence, Feasibility, Quality of Data

The Garmin Vivoactive5 was reported by the participants to be comfortable, and there were no adverse skin conditions due to continuous wearing of the smartwatch for up to 7 days. The primary nurses reported that the smartwatches were easy to charge (<2 h) and that there was no resistance from the participants to return the smartwatches afterwards. Feasibility for the placement of radar technology within the participants’ rooms, access to a power source and secure Wi-Fi, and 24–7 donning of the smartwatches was good, based on the researcher’s experience with these factors and adherence to protocols by the participants and healthcare providers. An initial data extraction of the first and last partial days of the acceleration data (days when the devices were donned and doffed by the researcher) was performed, based on the established daily profile threshold of at least 21 h of acceleration data per day. Adherence, measured by the total wear time (Garmin Vivoactive5) calculated after this initial data extraction of the first and last day, was 96% (Figure 1).

After the initial data extraction of the first and last partial days of acceleration data and the inspection of the remaining days of acceleration data and sleep digital biomarkers’ quality post retrieval, we obtained a total of 42-day/night data pairs (84 total data observations per digital biomarker analysis) that conformed to the established threshold of 21 available hours and at least two hours of nighttime data. The digital sleep variables collected included four digital biomarkers and 168 total observations (42 nights × 4) included in the final analysis of the digital sleep variables. Final analysis for all digital parameters, therefore, included a total of ten participants with complete (21 h minimum) digital profiles and included 42 total day/night pairs. Conversely, it should be noted that the correlations explored between the digital biomarkers and traditional outcome measures required a comparison of an equal number of observations, resulting in the use of the mean of the total sum of each digital biomarker, and in twenty total data observations per correlation result. Comparability of the acceleration’s raw data was moderate (requiring one method of calculation) and was determined based on the amount of conversion needed to deliver a comparable universal acceleration metric (i.e., ENMO/MAD). In comparison, raw data collected via the radar technology had a high level of comparability, requiring no conversion post retrieval, and was taken directly from the Somnofy base data to be used for analysis. There was missing data at the baseline for the following reasons: death or withdrawal from the study, periodic removal of the smartwatch by the participant, removal of the smartwatch by the healthcare provider for charging, participant not sleeping in their bed (i.e., sofa or chair in the room or in another room), behavioral and psychological activities (i.e., wandering, frequent awakenings), or technology recording or transfer failure due to the means of retrieving raw data (data broker).

### 4.2. Digital Snapshot: Day/Nighttime Physical Activity and Sleep Quality

A digital snapshot (Figure 2) is provided for the reader as a visual illustration of the digital combination of day/nighttime physical activity and sleep variables that were captured using Garmin Vivoactive5 and Somnofy. The participant was a 79-year-old male with diagnoses of diabetes, heart disease, and dementia (CDR = 2). The participant was taking twelve total medications, and two that were prescribed nightly for sleep disturbance. He scored ≥6/12 for behavioral symptoms (NPI-NH) of delusion [6], agitation/aggression [6], irritability [8], and nighttime behaviors [6]. His baseline PSMS score (10/30) suggests that minimum assistance is needed for activities for daily living; however, the mean daytime physical activity score of 34.6 milli-g (ENMO) indicates extreme sedentary daytime behaviors. We also note that the participant was taking a total of twelve daily medications, including two with psychotropic properties for sleep disturbance.

### 4.3. Digital Biomarker Comparisons

ENMO and WASO demonstrated an inverse relationship (Spearman correlation coefficient −0.34; *p* = 0.03), suggesting that higher daytime activity levels were connected to decreased nightly awake time (Table 3, Figure 3). Moreover, the correlation between the mean amplitude deviation (MAD) and the sleep regularity index (SRI) (0.43, *p* = 0.01) indicates that increased daytime activity improves sleep regularity (Table 3, Figure 3). These results highlight the importance of accurate monitoring and timely intervention for improved physical activity levels, as moderate levels of activity indicated superior sleep quality. Finally, the correlation between SE and WASO (−0.69, *p* < 0.000) (Table 3) confirms the previously suggested inverse relationship between these two digital biomarkers and showed that a higher SE is related to deceased wakefulness at night. We would like to reiterate to the reader that the results derived between the digital biomarkers utilized a total of 42 day/night pairs (84 total data observations). However, given the small sample size, these results should be interpreted as preliminary and exploratory. A complete list of all correlations that were explored is provided for the reader in Table 3.

### 4.4. Traditional Outcome Measures and Digital Biomarkers

A higher sleep score (0.70, *p* < 0.000) was related to a higher dementia rating on the CDR, suggesting that this digital biomarker improved for participants in later stages of dementia (Table 3, Figure 3). The NPI-NH-K (sleep behaviors) scores were related to both total (0.68, *p* = 0.02) and psychotropic medication (0.82, *p* < 0.000) usage (Table 3). NPI-NH-K also correlated with results from the digital biomarker WASO (0.58, *p* = 0.03), suggesting that WASO may be promising for enhancement of current proxy-rated assessment tools (Table 3).

## 5. Discussion

This proof-of-concept study finds that the use of sensing technologies is feasible to demonstrate detailed changes in activity and sleep quality for people with dementia living in nursing homes. Adherence to use of the sensing technologies was high and the chosen sensing technologies were well-tolerated by the participants. Data quality was acceptable and missing data due to compatibility issues, lack of clarity, or technical failures was minimal. We found a correlation between the traditional measure NPI-NH (Section K nighttime behaviors) and WASO, indicating that WASO could be a promising digital tool, supplementing proxy-rated measures. This may provide a pathway for further investigation of the potential bidirectional relationship between physical activity and sleep quality and improve dosage recommendations for daytime activity levels. We were able to demonstrate, using two unique sensing technologies, that there are correlations between daytime physical activity levels and sleep quality indicators: WASO and SRI. The results imply that the quantity of physical activity drives change in sleep quality and that improvement in one indicator could concomitantly affect the other. These digital biomarkers may be promising clinical tools in identifying subtle changes in activity levels and sleep behaviors and support evidence of a bidirectional relationship between daytime activity and sleep quality. On the other hand, the study found a correlation between the dementia rating scores (CDR) and the sleep quality digital biomarker, sleep score, which is suggestive of improved sleep quality with higher dementia stages, and contradictory to previous research results. This particular result could reflect the low sample size; however, it also highlights a potential risk for reliance on digital biomarkers alone in assessing sleep quality. Adding to this, the number of total and psychotropic medications were correlated to the NPI-NH-K scores, with higher scores being suggestive of increased polypharmacy and nightly psychotropic medication use.

Because inactivity and sleep disturbances are prevalent among people with degenerative diseases [36,37,38], improved sleep quality and activity are targetable clinical goals for this population. Most previous findings regarding the relationship between physical activity and sleep quality are inconsistent, and the tools used within previous studies to measure activity levels and sleep disturbance have mostly consisted of low validity proxy-rated questionnaires [39]. A meta-analysis by Atoui et al. [39] found that three variables were associated with activity levels the following day: sleep quality, SE, and WASO. However, results were high in variability and showed no association at an individual level. The bidirectional association between activity levels and sleep quality at an individual level has been studied [40] longitudinally, using gold-standard polysomnography (3 nights) and accelerometry (7 days) in healthy older adults (age 65–86 years, N = 92). The study found that low-intensity physical activity had a positive effect on sleep in older adults, specifically deep sleep stages, and the results suggest that bidirectional associations exist between sedentary behavior and sleep quality, although they remain ambiguous [40]. The association between sedentary behavior, physical activity, and sleep was also assessed using accelerometry (4 days) in healthy older adults (N = 157, >60 years old), and the results again emphasize the effect of daytime inactivity on sleep quality, concluding that moderate levels of activity were associated with improved sleep-quality parameters (TST and WASO) [41].

Previous studies report that sleep disturbances can be precursors to the prediction of Alzheimer’s disease and that sleep quality is decreased with advancing stages of cognitive impairment [42]. However, most of these previous observations are based upon cross-sectional periods using proxy-rated questionnaires, and with minimal participants in advanced stages of Alzheimer’s disease or dementia [42]. There is limited evidence as to the development of digital biomarkers, such as sleep score, SE, and WASO, regarding advancing dementia stages. People with dementia residing in nursing homes often present with heterogeneity and complex medical/pharmaceutical profiles. The results of our study indicated that a higher stage of dementia (CDR rating) was associated with an increasing or improved sleep score. To place this finding in perspective, we must highlight that polypharmacy use within the group was high, ranging from 2 to 13 prescribed daily medications, and included an average of two nightly psychotropic medications indicated to be for sleep disturbance. The interactions and half-life of these medications should be considered when evaluating sleep quality for this population. Psychotropics may improve sleep for people with dementia; however, benefits vary, and risks are significant [43], which supports the current recommendations for minimal use of these types of drugs for sleep disturbances. We believe that this study’s results, connected to perceived improvements in sleep quality in participants with moderate-to-severe dementia (CDR ≥ 2), are related to polypharmacy and nightly psychotropic medication use, which is further supported by the observed concomitant increase in CDR scores and medication usage within the study’s cohort. Higher NPI-NH-K scores, indicating greater sleep disturbance, were also associated with an increased use of total medications and psychotropics within the study cohort. Additionally, the digital biomarkers used within this study are constrained by the assumptions underlying their calculations. For example, the calculation for SE includes total sleep time, which is found by subtracting awakenings after sleep onset. It is possible that within this study, the use of psychotropic medications improved digital biomarkers by decreasing or improving the amount of time that the participants were awake after falling asleep. This theoretical improvement in sleep quality metrics is an indication of the limitations of using digital biomarkers alone for assessment of sleep quality in this complex population. A multifactorial sleep quality assessment should be conducted and include a blending of caregiver and physician feedback, traditional outcome measures, and promising new digital biomarkers. Despite the sample size within this study, this could be an important finding and should be explored longitudinally in a larger study, in connection with polypharmacy use and sleep disturbances in people with dementia residing in nursing homes.

Within this study’s group profile, some participants had relatives that visited several times a week, offering additional opportunities for increased activity, highlighting the potential influence of the informal carer in decreasing inactivity within institutional settings [44]. The appropriate dosage of activity for people with dementia living in a nursing home is less investigated and digital biomarkers could be a valuable tool in the future development of recommended activity levels for this population. Despite the routine environment afforded by schedules in the nursing home, the participants showed a high degree of daily variability in their physical activity levels and sleep quality. This suggests that a combination of proxy-rated measures and digital biomarkers, including both group-level and idiographic analysis, would be most appropriate for providing tailored future assessments.

This study reports results for two universal acceleration metrics, ENMO and MAD, and exposes some key differences in these techniques when blending digital biomarkers. Previous studies using these methods showed acceptable agreement between the results, using ENMO and MAD for the transformation of raw accelerometry data (using different sensing devices), and for the purposes of classifying active and sedentary activities [45,46]. However, to our knowledge, there are no studies which cross-analyze these universal metrics with other digital biomarkers, such as the defined sleep quality variables in this study. It can be that the differences within our results using these two methods are exacerbated by the complex combination of data representing several contexts: day vs. night, activity vs. sleep, acceleration vs. radar data. The mean daytime activity levels (ENMO and MAD) within the study highlight the high level of inactivity within the group. These totals fall within examples of established thresholds for ENMO and MAD (range = 9.3–118 milli-g), using single wrist-worn accelerometry to describe sedentary and light intensity activities: for example, lying down (9.3 milli-g, ±0.5), standing or sitting still (10.3 milli-g, ±1.6), washing pots (53.8 milli-g, ±5.4), and self-paced walking (103.1 milli-g, ±7.4) [47]. Differences between the discrepancies and sensitivities in these two methods for raw acceleration data translation should be further explored, as it could be that one method is more suitable for more complex evaluations (such as multiple day indexes). This should be further investigated to improve the generalizability of future studies by exploring the combination of multi-modal sensing technologies and traditional proxy-rated tools.

The questionnaires used in this study for the measurement of activity levels and sleep disturbance, PSMS and NPI-NH, are commonly utilized, proxy-rated tools in nursing home environments. Although the NPI-NH features a section dedicated to sleep, it does not specifically address the same sleep quality constructs as the digital measurements. We therefore chose a comparison of specific digital biomarkers, such as WASO, corresponding to the specific topics within the “nighttime behaviors” section of the NPI-NH, which is the most comparable to the digital constructs designed to highlight frequent awakenings. Similarly, the PSMS measures six functional areas related to activities of daily living, compared to acceleration data that precisely measures activity levels and their frequency. To better define and illustrate physical activity levels captured by the PSMS, this study chose to use a global score with assigned activity/functional levels. Other traditional tools not included in this study may offer a superior blending of constructs for the future use of sensing techniques in clinical practice and should be explored in future studies.

Ethical implications for the use of digital biomarkers and knowledge-based machine learning in healthcare decision-making for people with dementia living in a nursing home should be considered [48]. Challenges include inherent bias, data privacy, depersonalization of patient care, and algorithmic and dataset transparency [49]. For vulnerable groups of patients, such as people with dementia, integration of standardized protocols and regulatory frameworks should be developed, decreasing the substantial variation in metrics and models which complicate the generalizability and replication of interventions [49]. These frameworks must be adaptable to the pace of rapid development of innovative technologies and algorithms.

### Limitations

This study has several limitations. First, we note that although people with dementia can typically experience large variations in their daily activity and sleep behaviors, depending on the severity of their diagnosis, the nursing home environment provides a certain stability in daily routines, including planned activities, prescription administration, meal service, and sleep schedules. Secondly, the sensor number, placement of the smartwatches on the wrist, physiological differences in movement with age, specific diagnosis, polypharmacy profiles, and functional assistance level (i.e., ambulation with or without an assistive device) all contribute to the quality of the activity and sleep analyses.

The sample size used in this study is consistent with the design for a pilot study (DIPH.DEM) focused on the process and management aspects of informing a larger trial; however, because of the relatively low sample size, results should be interpreted with caution by the reader. Descriptive and statistical analysis are provided to improve the generalizability of results. Additionally, we note that sample size relativity should be considered when using digital biomarkers for analysis, and it is recommended that future studies combine self-reported and device-measured data, utilizing nomothetic and idiographic statistical analysis and conducting longitudinal observations, including >50 time points or days of observation [15]. This study uses 42 day and night pairs (84 total observation points), exceeding the recommendations for the amount of time points and observations. This proof-of-concept study also used high thresholds (21/24 h digital profile) for data inclusion and quality, to improve the generalizability of the results. Further, the inclusion of longitudinal data would provide a richer analysis of the changes in daytime activities and sleep quality over time, and we note that this is the first paper in a series, and that ongoing data collection up to one year from the baseline will be included in future articles.

The included sleep score algorithm used for analysis was proprietary; however, Somnofy provided the following explanation: “The sleep score investigates whether the participant has had enough sleep (duration), including quantity within each sleep stage. This is based upon general recommendation and previous research on the population, adjusted for sex and age. The score is reduced dependent on how much fragmentation is seen based on these norms.” Therefore, due to the black box nature of this algorithm, it is reported for the reader as a reference only. Another limitation of the study is that, although validated within other populations, the chosen sensing technologies were not specifically validated in the population of interest. Lastly, and perhaps most importantly, the lack of standardized metrics and complimentary statistical methods within the field of HAR is a limitation for the generalizability of the results. To improve the generalizability, this study follows examples from similar previous studies [45,46,47] that used common universal metrics and statistical analyses for translation of the raw data.

## 6. Conclusions

The results of this study show that the use of sensing technologies for monitoring activities and sleep quality in people with dementia is feasible in a nursing home environment. This study confirms that there is a relationship between daytime activity and sleep quality; however, the bidirectional relationship remains ambiguous and should be further investigated. Sensing technologies and use of digital biomarkers has the potential to identify subtle changes in physical activity levels and sleep quality, improving clinical assessment and the corresponding care recommendations.

## Figures and Tables

**Figure 1 sensors-25-06635-f001:**
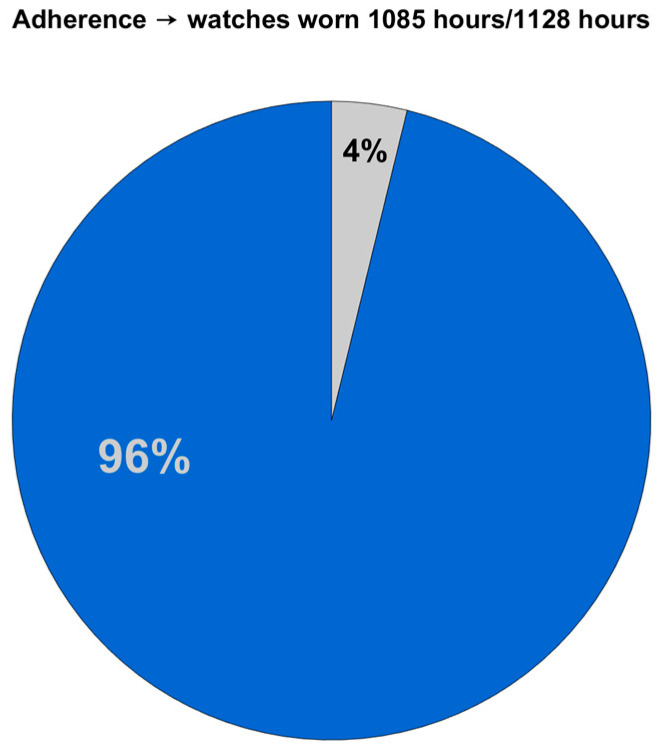
Total adherence (total wear time) to Garmin Vivoactive5, based on 5 days acceleration data per participant (excludes first and last day of data collection).

**Figure 2 sensors-25-06635-f002:**
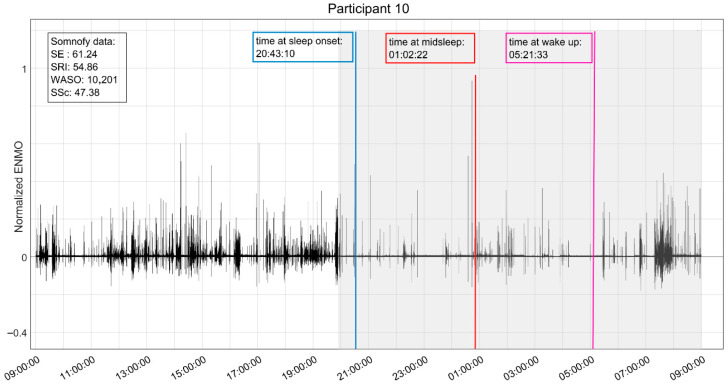
Combined 24 h physical activity from Garmin VivoActive5 (ENMO), total time in bed (gray background), time at sleep onset (blue line), time at midsleep (red line) time at wake up (pink line), and corresponding mean sleep quality digital biomarkers from Somnofy; sleep efficiency (SE), wake after sleep onset (WASO), sleep score (SSc), and sleep regulatory index (SRI).

**Figure 3 sensors-25-06635-f003:**
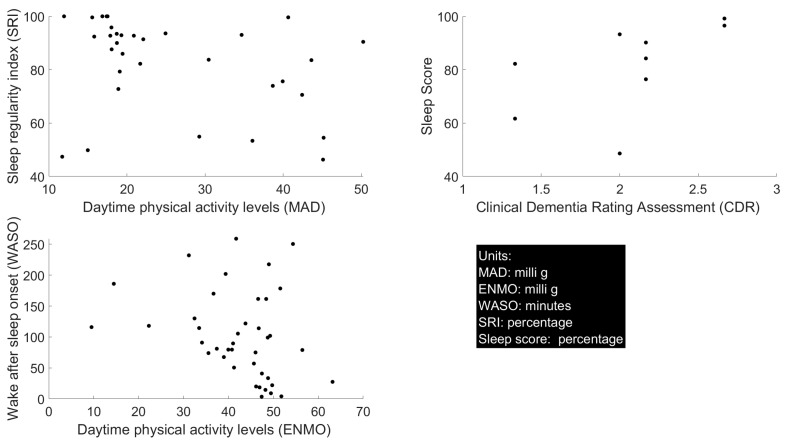
Relationships between daytime physical activity (acceleration) measured with ENMO/MAD vs. sleep quality digital biomarkers: WASO and SRI (left); relationship between digital biomarker sleep score and traditional measure CDR (top-right); unit description for each variable (bottom-right).

**Table 1 sensors-25-06635-t001:** Sample characteristics (N = 11).

Variables	Mean, Standard Deviation, Range (N)
Age, years	84.0 ± 6.1, 77–93 (11)
Gender, female (male)	8 (3)
Months in the nursing home	17.6 ± 12.5, 3–42 (11)
Dementia as primary diagnosis	6
Alzheimer’s dementia	3
Parkinson’s disease	1
CDR score (0–3)	2.2 ± 0.7, 1–3 (11)
Mild	4
Moderate	5
Severe	2
PSMS score (6–30)	15.7 ± 5.7, 8–23 (11)
Comorbidities (GMHR)	2.6 ± 0.7, 1–3 (11)
NPI-NH	
Agitation	2.2 ± 2.2, 1–6 (5)
Delusion	3.0 ± 2.4, 1–6 (4)
Hallucinations	3.0 ± 2.6, 1–6 (3)
Depression	2.0 ± 1.4, 1–4 (4)
Anxiety	1.3 ± 0.6, 1–2 (3)
Apathy	5.8 ± 4.6, 1–12 (4)
Irritability	3.6 ± 2.9, 1–8 (5)
Euphoria	8.0 ± 0.0, 8–8 (1)
Disinhibition	1.5 ± 0.6, 1–2 (4)
Aberrant motor behavior	1.0 ± 0.0, 1–1 (4)
Nighttime behavior disturbances	4.7 ± 2.9, 1–8 (6)
Appetite/eating	4.0 ± 2.9, 1–8 (4)
Total medication	7.6 ± 3.4, 2–13 (11)
Psychotropic drugs	1.9 ± 1.4, 1–5 (8)

Abbreviations: CDR: Clinical Dementia Rating, GMHR: General Medical Health Rating, NPI-NH: Neuropsychiatric Inventory–Nursing Home version, PSMS: Personal Self Maintenance Scale.

**Table 2 sensors-25-06635-t002:** Results from selected digital biomarkers for physical activity and sleep quality indicators (N = 10); baseline data.

Variable	Mean ± SD	Range
Daytime activity (ENMO) *	42.5 ± 10.4	9.5–63.2
24 h activity (ENMO)	43.5 ± 9.2	17.0–65.6
Daytime activity (MAD)	26.5 ± 10.5	11.7–47.2
M5 (5 h with highest activity)	52.2 ± 11.7	15.3–69.2
Sleep efficiency (SE) (%)	70.8 ± 17.9	15.4–94.8
Sleep score Somnofy (%) †	76.4 ± 25.3	11.6–100.0
Wake after sleep onset (WASO), min	102.5 ± 70.6	3.5–258.9
Sleep regularity index (SRI, %)	72.1 ± 9.8	53.0–92.8
Total sleep time (TST), h	9.6 ± 2.8	1.37–18.1
Sleep latency, min	35.6 ± 48.1	0–244.9
Total time in bed, h	10.6 ± 2.7	1.9–18.6
No presence, min	7.7 ± 9.8	0–47.5

* Analysis performed using daytime metrics, † Sleep score is reported as a reference only (proprietary algorithm). Abbreviations: ENMO, Euclidean norm minus one; MAD, magnitude amplitude deviation; M5, mean of the five most active hours of the day (milli-g); WASO, wake after sleep onset; TST, total sleep time; SRI, sleep regularity index.

**Table 3 sensors-25-06635-t003:** Spearman (rho) correlations (N = 10) with bias-corrected affected bootstrapping (BCa) including standard error, *p*-value, and 95% confidence interval (CI).

Explored Relationships	ρ (Spearman)	Std Error	*p*-Value	95% CI
CDR–SE	0.41	0.38	0.30	[−0.33, 1.1]
CDR–WASO	−0.35	0.33	0.30	[−1.0, −0.30]
CDR–Sleep Score	0.70	0.25	0.005 *	[0.21, 1.1]
CDR–SRI	0.14	0.36	0.70	[−0.56, 0.83]
ENMO day–SE	0.21	0.17	0.20	[−0.11, 0.54]
ENMO day–WASO	−0.34	0.16	0.03 *	[−0.65, −0.3]
ENMO day–Sleep Score	0.15	0.19	0.45	[−0.24, 0.53]
ENMO day–SRI	0.19	0.17	0.27	[−0.49, 0.52]
MAD day–SE	−0.01	0.18	0.58	[−0.44, 0.25]
MAD day–WASO	0.23	0.18	0.20	[−0.12, 0.58]
MAD day–Sleep Score	−0.39	0.20	0.05	[−0.78, 0.0]
MAD day–SRI	0.43	0.17	0.01 *	[0.09, 0.76]
NPI-NH-K–SE	−0.48	0.34	0.16	[−1.14, 0.18]
NPI-NH-K–WASO	0.58	0.26	0.03 *	[0.07, 1.1]
NPI-NH-K–Sleep Score	−0.17	0.42	0.69	[−1, 0.66]
NPI-NH-K–SRI	−0.19	0.39	0.63	[−1.0, 0.58]
NPI-NH-K–Total meds	0.68	0.28	0.02 *	[0.13, 1.24]
NPI-NH-K–Psychotropics	0.82	0.09	0.000 *	[0.63, 0.99]
PSMS–ENMO day	−0.11	0.33	0.74	[−0.33, 0.31]
PSMS–MAD day	0.10	0.41	0.80	[−0.76, 0.54]
SE-WASO	−0.69	0.10	0.000 *	[−0.87, −0.50]

* *p*-value < 0.05. Abbreviations: CDR: Clinical Dementia Rating scale, ENMO: Euclidean norm minus one, NPI-NH (Section K): neuropsychiatric inventory (nighttime behaviors), MAD: mean amplitude deviation, meds; medications, PSMS: Personal Self-Maintenance Scale, Psychotropics; nightly medication prescribed for sleep disturbances, SE: sleep efficiency, SRI: sleep regulatory index, WASO: wake after sleep onset.

## Data Availability

All data supporting the results and conclusions of this article will be made available by the authors, without undue reservation.

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
