# Peer review of "Use of Digital Biomarkers from Sensing Technologies to Explore the Relationship Between Daytime Activity Levels and Sleep Quality in Nursing Home Residents with Dementia: A Proof-of-Concept Study"

_sensors, 2025, doi:10.3390/s25216635_

Round 1

Reviewer 1 Report

Comments and Suggestions for Authors

This is a well-written paper exploring the application of a wearable device and a home-based monitoring device to assess sleep patterns and sleep quality in nursing home residents. The novelty is sound and the approach has clear potential for broader application.

  1. Although the authors indicate that this is a proof-of-concept study, the sample size is relatively small. As shown in Tables 1 and 2, there is substantial heterogeneity across participants, and the total wear time is limited. Forty day/night pairs may be too few to draw firm conclusions. It would be helpful if the authors could also provide 95% confidence intervals for the reported Spearman coefficients to better convey the uncertainty around the estimates.
  2. Because the participants had dementia and are more likely to experience sleep disruption, it would be valuable to explore whether there is any association between dementia severity and sleep quality metrics.
  3. The presentation of results could be improved. A consolidated table or a heatmap summarizing the correlations of all comparison pairs would help readers more easily interpret the outcomes.
  4. Although PSMS was measured, the association between PSMS and device-derived outcomes is not presented, and is only briefly mentioned in the discussion. It would again be helpful if the authors could report all outcomes in a consolidated table or heatmap to make these relationships clearer.
  5. Neither the Garmin nor the Somnofy devices have been validated in this specific population. The authors may wish to briefly acknowledge this as a limitation in the discussion section.

Author Response

Comment: Although the authors indicate that this is a proof-of-concept study, the sample size is relatively small. As shown in Tables 1 and 2, there is substantial heterogeneity across participants, and the total wear time is limited. Forty day/night pairs may be too few to draw firm conclusions. It would be helpful if the authors could also provide 95% confidence intervals for the reported Spearman coefficients to better convey the uncertainty around the estimates.

Response: Thank you for your comments and suggestions. First, the suggestion for the additional confidence intervals is greatly appreciated and we believe this is a wonderful addition to the analysis and to the eventual publication. Upon closer inspection of the data through the process of recalculating the results based on your recommendation to include these new analyses, we were able to increase the amount of available data for analysis of the digital biomarkers from 40 to 42 day-night pairs (84 total observations). You will see this slight increase reflected within the results section. We have added additionally a bias tool and bootstrapping to the Spearman analysis, now highlighted in the “Statistical Analysis” section lines 381-384, to strengthen the analyses based upon the low sample size. This did change one of our findings (SE and daytime activity), which we have now corrected within the abstract and results. We now provide the Spearman coefficient (rho), standard error, p-value, and confidence intervals for all explored relationships between variables (reference the now included Table 3 under results). Secondly, based on both reviewers’ comments we realize that we have not explained the methodology thoroughly enough to emphasize the reason for the low sample size and further the number of total observations (digital biomarkers) for each analysis within the sample. The primary study, DIPH.DEM, is a pilot study for a larger project. The overarching methodology of DIPH.DEM and sample size estimates are now further explained in lines 146-171.  We feel this was a positive and necessary addition and thank you for the opportunity to clarify this information for the reader. The total wear time for the watches was 7 continuous days, however adherence was calculated after the exclusion of the first and last partial days (the days the devices were donned and doffed by the researcher as these represented only a half day of data). This is further explained in lines 273-276, and again in results lines 475-481 (Figure 1 added for adherence). Seven days is a standard amount of total recording time according to previous literature and the most common observation times using these types of sensing technologies in previous literature are 24 hours followed by studies using 3-7 days in duration (reference Boyle et al. umbrella review in Sensors). Additionally, the analysis of 42 day-night pairs (84 total observations) exceeds the recommendations for observation points (>50 observations) needed for analysis of digital biomarkers (see lines 150-170). Observation time prior to extraction of missing or incomplete data for all sleep data was 6 nights (21:00-06:59). We include a total of 42 nights and 168 observations for sleep analysis using the digital biomarkers (lines 493-508). Again, we thank you for the chance to clarify this information.

Comment: Because the participants had dementia and are more likely to experience sleep disruption, it would be valuable to explore whether there is any association between dementia severity and sleep quality metrics.

Response: Thank you for this helpful and thoughtful suggestion. We believe this will add valuable insight to the analysis in the article. We have now included a Spearman correlation coefficient analysis between the dementia severity scores from the Clinical Dementia Rating scale (CDR) in Table 3, which now clearly lists all explored correlations. The results are included in lines 585-593 under the revised results section 4.5 titled Traditional outcome measures and digital biomarkers. Additionally, we include a paragraph in the discussion to address these important new observations. Your suggestion also inspired a deeper look into the use of total and psychotropic medications in relation to the dementia stage within the group, and further description of the medication use related to the traditional sleep assessment tool. This information is now included in Table 3 and expanded upon in the discussion (lines 679-730). We feel these new additions based on your recommendation provide a more holistic view of the complex relationship between sleep quality, dementia rating, and polypharmacy within the group. Thank you again for this valuable feedback.  

Comment: The presentation of results could be improved. A consolidated table or a heatmap summarizing the correlations of all comparison pairs would help readers more easily interpret the outcomes.

Response: Thank you for the suggestion for an improved presentation of the correlation results and we agree with your recommendation. We have now included a new summary within the results section (Table 3) of all explored relationships/correlations for the reader, which we hope will provide clearer understanding of the results.

Comment: Although PSMS was measured, the association between PSMS and device-derived outcomes is not presented, and is only briefly mentioned in the discussion. It would again be helpful if the authors could report all outcomes in a consolidated table or heatmap to make these relationships clearer.

Response: Thank you for this comment. We will kindly refer to the previous response, which states that all correlation results are now presented in Table 3.

Comment: Neither the Garmin nor the Somnofy devices have been validated in this specific population. The authors may wish to briefly acknowledge this as a limitation in the discussion section.

Response: We agree that this is an important point. We have added a statement in the discussion regarding the validation of the Garmin and Somnofy within this specific population under the limitations section, lines 829-831.

Reviewer 2 Report

Comments and Suggestions for Authors

In the manuscript titled " Use of Digital Biomarkers from Sensing Technologies to Explore the Relationship Between Daytime Activity Levels and Sleep Quality in Nursing Home Residents with Dementia: A Proof-of-Concept Study″ the authors present the relationship between daytime activity and sleep quality. However, the study needs to be supplemented, and its key parts expanded to make it complete if it is to be published in the Sensors journal.

  1. In the Abstract, it is emphasised that the authors investigated the relationship feasibility and adherence to the use of sensor technology. Can you explain how this was done? The results in the manuscript show the relationship between daytime activity and sleep quality, but the results regarding the feasibility and adherence of the use of sensor technology are not presented independently. Also, can you explain why is the number of female participants emphasised in the abstract section?
  2. The Introduction section is very well written, logical and comprehensive. However, some references (lines 77 and 82) were not included in the text correctly, so this should be corrected.
  3. The impartiality of data collection should be explained in greater detail.
  4. The number of participants (N=11) is very low, and it should be noted that the patients were diagnosed with dementia or moderate to severe cognitive impairment assessed by the Clinical Dementia Rating Assessment (CDR) (line 111). However, this issue should not only be considered as a limitation, as mentioned in the Limitations section (lines 464 - 473), but also recognised as a major factor affecting the validity and interpretation of the obtained results. As explained, the number of day and night pairs is below the required threshold, which should be improved. Moreover, the results were collected for a relatively short period of time of only three months (line 271). The main concern of the study, therefore, lies in the insufficient amount of data and heterogeneity of the examined patient population. Namely, given that only 11 participants were included, their CD score (as shown in Table 1) varies widely. This heterogeneity—ranging from mild to severe cognitive impairment (CDR scores between 0.5 and 3)—could substantially influence the study outcomes and compromise the reliability of the conclusions. To enhance the robustness and interpretability of the results, the patient cohort should be more homogeneous and include a larger number of participants. A reconsideration of the study population is strongly recommended.

Author Response

Comment: In the Abstract, it is emphasised that the authors investigated the relationship feasibility and adherence to the use of sensor technology. Can you explain how this was done? The results in the manuscript show the relationship between daytime activity and sleep quality, but the results regarding the feasibility and adherence of the use of sensor technology are not presented independently. Also, can you explain why is the number of female participants emphasised in the abstract section?

Response: This was a helpful observation, and we thank you for your feedback. We have expanded upon the paragraph describing feasibility and data processing for calculation of adherence within the study in lines 273-276 and again in the results lines 475-481 (reference new figure 1 for adherence). The number of female participants has been removed from the abstract based upon your feedback. This was originally included in this format as a reflection of the way we reported the results in Table 1. We thank you again for these helpful suggestions.

Comment: The Introduction section is very well written, logical and comprehensive. However, some references (lines 77 and 82) were not included in the text correctly, so this should be corrected.

Response: Thank you for this comment. The references mentioned have now been amended.

Comment: The impartiality of data collection should be explained in greater detail. **(How we collected the data – impartially without bias/description of data collection)

Response: This is an important observation, and we have now included further details about the impartiality of the data collection under the methods section of the paper, lines 146-182.

Comment: The number of participants (N=11) is very low, and it should be noted that the patients were diagnosed with dementia or moderate to severe cognitive impairment assessed by the Clinical Dementia Rating Assessment (CDR) (line 111). However, this issue should not only be considered as a limitation, as mentioned in the Limitations section (lines 464 - 473), but also recognised as a major factor affecting the validity and interpretation of the obtained results. As explained, the number of day and night pairs is below the required threshold, which should be improved. Moreover, the results were collected for a relatively short period of time of only three months (line 271). The main concern of the study, therefore, lies in the insufficient amount of data and heterogeneity of the examined patient population. Namely, given that only 11 participants were included, their CD score (as shown in Table 1) varies widely. This heterogeneity—ranging from mild to severe cognitive impairment (CDR scores between 0.5 and 3)—could substantially influence the study outcomes and compromise the reliability of the conclusions. To enhance the robustness and interpretability of the results, the patient cohort should be more homogeneous and include a larger number of participants. A reconsideration of the study population is strongly recommended.

Response: Thank you for your helpful suggestions. The sample size was mentioned by both reviewers, and we realized that we had not properly clarified the reason for the chosen methods for the reader. This is now included under methodology section, lines 146-182. We have, as suggested by the first reviewer, now revised the statistical analysis (standard error, confidence intervals, bootstrapping with bias control Spearman correlations) to provide more information about the confidence of results and to improve generalizability (see the added Table 3 under results). Additionally, we have added statements in the methodology (383-388), results (576-581), and limitation section (803-807) that recommend caution for interpretation of the results based upon the sample size. We further try to more clearly describe the amount of data that was used within the sample to analyze the digital biomarkers. The data observations exceed the recommended amount for this type of analysis, which we now describe further in lines 154-170 under the methods section. The CDR scores were between 1-3 with a mean of 2.2 within the group, however we understand why this was misunderstood previously. This has now been corrected in the results section in lines 398-421. Further, we have now included information about the relationship between the dementia stage and sleep quality variables, as recommended by the first reviewer. We believe this adds a more holistic view of the complex relationship between dementia stage, polypharmacy use, and sleep quality. You will see this addition in Table 3 and in lines 585-592, and the results further expanded upon within the discussion (lines 679-730). We agree with you that a larger sample size and a longitudinal design would enrich the provided analysis. The DIPH.DEM study is an external pilot study to a larger trial (now further explained under the methodology section) with plans for extension of these analyses within the now ongoing larger study. DIPH.DEM’s intended aim was to provide information for the larger study regarding the chosen methodological process, the feasibility of the use of these sensing techniques for measuring activity and behavioral changes, and proof of concept for use of digital biomarkers as a complementary assessment tool for people with dementia living in a nursing home. We hope that the further description we have provided will give more meaning to the methodology and sample size surrounding this study. We again thank you for your thoughtful recommendations.

Round 2

Reviewer 2 Report

Comments and Suggestions for Authors

After implementing extensive revisions to the manuscript and emphasizing that this work represents a pilot study in all its aspects, including the interpretation of results and the discussion, this version constitutes a substantially different manuscript and should be interpreted accordingly.

Author Response

Comment: After implementing extensive revisions to the manuscript and emphasizing
that this work represents a pilot study in all its aspects, including
the interpretation of results and the discussion, this version
constitutes a substantially different manuscript and should be
interpreted accordingly.

Some of the references (lines 77 and 82) were not included in the text
correctly, so this should be corrected

Response: Thank you for your response and for taking the time to review the revisions to the manuscript. We are grateful for this opportunity to improve the presentation for the reader. We have now checked that all references are included correctly and have made some minor changes regarding sentence structure to further improve clarity for the reader. We wish you a wonderful weekend.